# Dynamic Change of Aroma Components in *Chimonanthus praecox* Flower Scented Teas During Absorption and Storage

**DOI:** 10.3390/foods14101696

**Published:** 2025-05-11

**Authors:** Xiongyuan Si, Hao Zuo, Penghui Li, Ye Tan, Mangmang Tan, Zhihui Chen, Changsong Chen, Taolin Chen, Zhonghua Liu, Jian Zhao

**Affiliations:** 1Biotechnology Center, Anhui Agricultural University, Hefei 230036, China; xysi@ahau.edu.cn (X.S.); tanye1213@163.com (Y.T.); 2Key Laboratory of Tea Science of Ministry of Education, College of Horticulture, Hunan Agricultural University, Changsha 410128, China; zuohao0116@163.com (H.Z.); zhonghua-liu@hunau.edu.cn (Z.L.); 3State Key Laboratory of Tea Plant Biology and Utilization, Anhui Agricultural University, Hefei 230036, China; lphui2012@126.com (P.L.); tanmang07@163.com (M.T.); 4Tea Research Institute, Fujian Academy of Agricultural Sciences, Fuzhou 350013, China; chenzhihui75@sina.com (Z.C.); ccs6536597@163.com (C.C.); 5College of Tea Science, Guizhou University, Guiyang 550025, China; tlchen@gzu.edu.cn

**Keywords:** *Chimonanthus praecox* flowers, scented tea, aroma compounds, Chimonanthus tea, storage

## Abstract

Flower-scented teas become increasingly popular to new generations, due to their infused floral essences of diverse volatile compounds and additional health functions. Flower-scented teas have significantly broadened the spectrum of aroma perception, intensity, and longevity. Here, *Chimonanthus praecox* flowers were used to scent tea dhools to create different Chimonanthus teas with strong and characteristic aromas. The dynamic absorption of aromas by three tea dhools, and aroma compatibility in three flower-scented teas, and the aroma retention in Chimonanthus teas during storage were investigated. At least twelve aroma compounds were selectively absorbed by three tea dhools, with seven compounds, pulegone, 3-phenylpropanol, (E)-cinnamaldehyde, cinnamyl alcohol, γ-phenylpropyl acetate, (E)-isoeugenol, and (E)-cinnamyl acetate, commonly absorbed to three Chimonanthus teas. The different absorption preferences to floral volatiles and absorption capacity of three tea dhools could be related to their surface structures and trichome conditions. Linalool, phenylmethyl acetate, and methyl salicylate as significant volatile components were substantially enhanced for both Chimonanthus flowers and tea dhools, thereby augmenting the floral bouquet of Chimonanthus tea. After 56 days of storage, alcohol volatiles emerged as the predominant volatile types, although esters are the major contributors to the aroma of freshly prepared Chimonanthus teas.

## 1. Introduction

Tea represents the most widely consumed beverage globally, second only to water, for its pleasant flavors and numerous health benefits [1,2]. However, the flavor characters and health functions of the six major types of tea show significant differences, due to their distinct processing procedures and technologies. As one of the six types of tea, yellow tea can be distinguished from other types of tea, such as green tea, black tea, Oolong tea, Pu-erh tea, and white tea, by its special aroma [3]. Yellow tea is recognized for its elegant fragrance and smooth taste [3]. According to the plucking standards and tenderness of raw materials, yellow tea is divided into three categories, little yellow tea, bud yellow tea, and large-leaf yellow tea (LLYT). Although LLYT is consumed in smaller regions and its characteristic flavor and aroma are not equally popular to young generations as other types of tea, LLYT has received increasing attention because of its significant health benefits, such as reducing blood pressure and blood glucose levels [4,5]. In recent years, flower-scented tea has become increasingly popular in new generations, owing to its much more diverse and stronger characteristic aroma and additional health functions conferred by the herbal flowers [6,7,8,9]. Particularly, various innovative techniques on flower-scented tea have emerged to significantly enrich and diversify tea flavors while further pushing the conventional boundaries of tea processing. The successful Jasmine teas made from scenting green tea dhools with Jasmine (*Jasminum sambac*) flowers has a rich legacy [10]. Other flower-scented teas, such as Gardenia tea scented with *Gardenia jasminoides* flower and Osmanthus tea scented with *Osmanthus fragrans* flowers, have a great popular market due to their robust flavors and additional health benefits [6,7]. It is therefore desirable to produce flower-scented LLYT, in order to overcome the shortfalls of LLYT aroma and eventually improve its overall flavor quality, health functions, thereby its popularity and consumption.

*Chimonanthus praecox* plants are widely cultivated as ornamental trees in southern China, yielding yellow or pale yellow flowers that emit a strong and sweet fragrance during winter, a time when many other plants remain dormant [11]. The intense fragrances of *C. praecox* flowers, including terpenoids and benzenoids, exhibit anti-inflammatory, antioxidant, and antibacterial activities. Chimonanthus flowers have been harnessed to scent tea dhools, generating Chimonanthus tea with unique flavors and potential health benefits [12]. But whether Chimonanthus flowers are suitable to produce flower-scented LLYT with desirable aroma is not known.

It has been shown that fresh flowers from the same variety could exhibit varying effects on tea dhools and generate different aromas in the scented tea [12,13]. Linalool, methyl anthranilate, 4-hexanolide, 4-nonanolide, and (E)-2-hexenyl hexanoate have been recognized as contributors to the floral aroma of Jasmine tea [10]. By contrast, phenyl methanol, linalool, phenylmethyl acetate, (Z)-3-hexenyl benzoate, methyl anthranilate, indole, and *α*-farnesene were major volatile compounds in Jasmine tea [14]. These discrepancies may arise from variations in the Jasmine flowers, tea dhools, or processing techniques employed. Different tea dhools could also have different flower-scenting effects, due to different physical and chemical absorption mechanisms functioning in particular flower–tea dhool mixtures. For example, the predominant aromas in Osmanthus black tea encompass *β*-ionone, dihydro-*β*-ionone, phenylacetaldehyde, citral, geraniol, and linalool [15]. However, the key aromatic substances identified in Osmanthus green tea include linalool, *α*-ylangene, *α*-ionone, *γ*-decalactone, and 4-hydroxy-*β*-ionone [16]. The primary aromatic constituents of Osmanthus fragrans flowers, such as *β*-ionone, dihydro-*β*-ionone, and *α*-ionone, contribute to their floral and fruity bouquet [15].

In view of these variations and previous trials on Chimonanthus flower-scenting tea, with regard to sensory characters and storage properties, comprehensive comparisons of the effects of Chimonanthus flower scenting on different tea dhools are needed. These include LLYT for revealing the volatile absorption dynamics, storage properties concerning retention, and the longevity of adsorbed aroma in Chimonanthus flower-scented tea [8,12]. Given the robust health benefits of Chimonanthus flowers and the growing attention on LLYT health functions, we conducted a series of meticulous investigations into Chimonanthus tea preparations from LLYT as target tea dhool, Huangshan maofeng Tea (HSMF), and Dianhong jinzhen Tea (DHJZ) as representative tea dhools for referencing green tea and black tea, respectively [4,17]. This study encompassed the dynamics of fragrant volatile absorption among the three tea dhools during the scenting process, the interactions of volatiles from Chimonanthus flowers and tea dhools in various modalities and magnitudes, as well as the maintenance duration of volatiles in Chimonanthus tea. Simultaneously, this study also aims to enhance the health-promoting properties and the aromatic qualities of LLYT by scenting with *Chimonanthus praecox* flowers, thereby facilitating a greater acceptance of LLYT. Our study revealed distinctive characteristics of volatile interactions inherent in diverse tea dhool materials, and it has paved a road toward the improvement of tea processing technology to ensure a higher quality.

## 2. Materials and Methods

### 2.1. Chemicals and Regents

Sodium chloride (NaCl) was obtained from Sinopharm Chemical Reagent Co., Ltd. (Shanghai, China). Deionized water was prepared by a Milli-Q water purification system (Millipore, Billelrica, MA, USA). N-alkanes (C8-C40) were obtained from J&K Scientific (Beijing, China). N-hexanes (98%), and N-heptane (99%) were purchased from Shanghai Aladdin Biochemical Technology Co., Ltd. (Shanghai, China).

### 2.2. Collection of Tea and Chimonanthus Flowers Blooming

The high-grade Large-leaf Yellow Tea (LLYT) and Huangshan maofeng Tea (HSMF) used in this experiment were purchased from Xie Yuda Tea Co., Ltd. (Hefei, China) and Dianhong jinzhen Tea (DHJZ) from XinYiHao tea industry co., ltd. LLYT was processed according to national standards GB/T 39592-2020 (Code of practice for processing of yellow tea, Standards Press of China: Beijing, China, 2020), with a rough surface and loose structure. DHJZ was processed according to national standards GB/T 13738.2-2017 (Black tea—Part 2: Congou black tea, Standards Press of China: Beijing, China, 2017), with fluffy surfaces due to the presence of more trichomes. HSMF was processed according to national standards GB/T 19460-2008 (Product of geographical indication - Huangshan maofeng tea, Standards Press of China: Beijing, China, 2008), with rough surfaces and porous structures. The morphologies were observed by scanning electron microscopy (SEM, S-4800, Hitachi, Tokyo, Japan). *C. praecox* var. *Grandiflorus* blooming flowers in year 2023 winter were used in this experiment in Nongcui Garden of Anhui Agricultural University (Hefei City, Anhui Province, China; latitude: 31.55° N; longitude: 117.12° E) on 15 February 2023.

### 2.3. Processing of Chimonanthus Tea Samples

The scenting technology of Chimonanthus tea follows the method described by predecessors; that is, the processing method of two-time continuously scenting and one-time raising fragrance procedures is adopted, and appropriate modifications are made [12]. The details are as follows: mixing tea and flowers according to the mass ratio of 1:1, scenting at 25 °C for 20 h, and separating flowers from flowers after scenting, that is, completing the first scenting. Then, a second scenting was carried out under the same conditions. After the second scenting, the tea sample was dried in an oven at 75 °C for 45 min. Finally, the tea and flowers were mixed and scented for 12 h according to the mass ratio of 10:1, and the Chimonanthus tea sample was obtained. The tea and Chimonanthus tea samples used for HS-SPME-GC-MS analysis were ground to pass through 30 to 60 mesh for further use [18].

### 2.4. Extraction of Volatile Compounds by the Headspace Solid-Phase Microextraction

The headspace solid-phase microextraction (HS-SPME) was used for the collection of aromas from tea leaves and for GC-MS analysis, according to previously described methods, and appropriate modifications are made [12]. Briefly, about 1.0 g of tea or Chimonanthus tea was placed into a 20 mL sealed glass vial, and 3.2 g NaCl and 10 mL boiling distilled water was added. Tightly capped with a PTFE-silicon septum, the vial was immediately put into a thermostatic oscillator and incubated at 60 °C for 5 min with a stirring speed of 400 rpm. After 5 min of stabilization, the carboxen/polydimethylsiloxane/divinylbenzene(CAR/PDMS/DVB) coating fiber (Supelco, Inc., Bellefonte, PA, USA) was inserted into the headspace of the sample, extracting at 60 °C for 45 min with a stirring speed of 400 rpm. The aroma of Chimonanthus flowers was extracted with reference to the previous methods, and appropriate modifications were made [19]. About 1g of fresh Chimonanthus flowers were put into a 20 mL sealed glass vial, next, the vial was immediately put into a thermostatic oscillator and incubated at 25 °C for 5 min. Then, the SPME fiber was inserted into the headspace of the sample, extracting at 25 °C for 45min. All of the volatile compounds absorbed on the SPME fiber were desorbed at the GC-MS injector at 250 °C for 10 min and then immediately analyzed by GC-MS.

### 2.5. Qualitative and Quantitative Analysis of the Volatiles by GC-MS

The aromas of tea dhools, Chimonanthus flowers, and scented Chimonanthus teas were analyzed by Agilent 7890B gas chromatograph (GC, Santa Clara, CA, USA) equipped with a 7000B mass spectrometer (MS, Agilent, Santa Clara, CA, USA). The GC conditions were set as follows: The volatile compounds were separated on a DB-5MS capillary column (60 m × 0.25 mm i.d., 0.25 µm film thickness). An injection of 1.0 µL of the sample was used for the sample and standard volatile analysis. The sample injection temperature was set at 250 °C and helium (99.999%) was used as the carrier gas with a constant flow of 0.8 mL/min. The splitless injection mode was used. The temperature program was increased from 50 °C (hold 5 min) to 250 °C at 4 °C/min. The mass selective detector was operated in positive electron-ionization mode with a mass scan range from *m*/*z* 30 to 500 at 70 eV. The ion source temperature was set at 230 °C, and the transfer line temperature was at 280 °C [12]. All compounds were first identified by the National Institute of Standards and Technology (NIST 20) mass spectra library. And then, each compound was further confirmed by retention index (RI). The RI was calculated using the n-alkane series (C6-C40). The comparison of the calculated RI values with the reported RI values from the NIST Chemistry WebBook (https://webbook.nist.gov/chemistry/, accessed on 28 May 2024) database was performed for identification of volatile compounds. Based on the total ion current chromatograms, the relative content of each compound in a sample was calculated using the peak area normalization method [12].

### 2.6. Odor Description

The odor descriptions of aroma volatiles identified from *C. praecox* flowers, tea, and Chimonanthus flowers-scented tea refer to previous studies, and the online tool of Flavor Ingredient Library (https://www.femaflavor.org/, accessed on 15 January 2025), FlavorDB2 (https://cosylab.iiitd.edu.in/flavordb2/, accessed on 15 January 2025), Chemical Book (https://api.chemicalbook.com/, accessed on 18 January 2025), and Perflavory Information System (http://www.perflavory.com/search.php, accessed on 15 January 2025) were utilized for odor descriptions [3,20,21,22,23,24,25].

### 2.7. Statistical Analysis

Principal component analysis (PCA), hierarchical clustering analysis (HCA), and orthogonal partial least squares-discriminant analysis (OPLS-DA) were performed using SIMCA 14.1 (Umetrics, Umea, Sweden), with the normalized data scaled by Par. PCA and HCA were performed to evaluate the isolation effect between samples in an unsupervised mode. The VIP-scores of OPLS-DA were applied to evaluate the importance of detected features, and the reliability of the result was verified using a 200-times permutation test. A variable importance for the projection (VIP) > 1 in the OPLS-DA model served as a differential marker for the sample. Heatmap analysis was performed using MultiExperiment Viewer software (version 4.7.4, Oracle Corporation; Redwood, CA, USA). Venn diagrams were drawn on the venny 2.1 web server (https://bioinfogp.cnb.csic.es/tools/venny/index.html). GraphPad Prism 8 (GraphPad Software Inc., San Diego, CA, USA) and the online tool ChiPlot (https://www.chiplot.online) were used for plotting graphs. Origin 2024 (OriginLab Corporation, Northampton, MA, USA) was utilized for the flavor sankey diagram. All experiments in this work were performed in triplicate. The statistical significance was analyzed by analysis of one-way analysis of variance (ANOVA) via SPSS 19.0 Statistics software (Chicago, IL, USA) with a post hoc Tukey’s test at *p* < 0.05.

## 3. Results

### 3.1. Characteristic Composition of Volatile in Chimonanthus Flowers

Previous studies have shown that flowers from different *Chimonanthus praecox* varieties exhibit markedly distinct aroma profiles [12]. The impact of *C. praecox* var. *Concolor* on the aromatic components of tea dhools had been reported [12]. *C. praecox* var. *Grandiflorus* with yellow flowers and strong fragrance had not been used to scent tea dhools. We chose *C. praecox* var. *Grandiflorus* flowers for this study since this variety is also widely grown in southern China. Chimonanthus flowers typically bloom from December to February. The volatiles associated with Chimonanthus flowers are characterized by their sweet and pleasant aromas, making them suitable for use in various perfumes and flavor additives. The volatile compounds released by fresh Chimonanthus flowers were analyzed using headspace solid-phase microextraction combined with gas chromatography coupled with mass spectrometry (GC-MS). The characteristic chromatogram is presented in Figure 1A, while the content percentage of the 11 primary volatiles are detailed in Figure 1B. In total, 61 compounds were isolated and identified from fresh flowers, classified into seven categories: 7 alcohols, 2 aldehydes, 39 alkanes, 8 esters, 2 ketones, 1 nitrogen heterocyclic compound, and 2 phenolic compounds (Appendix A). The rose plot, which aggregates the content percentage of these seven categories, illustrates that alcohols and esters constitute the predominant components of Chimonanthus flowers volatiles, representing approximately 52% and 37% of the total volatiles, respectively (Figure 1C). Notably, linalool, phenylmethyl acetate and methyl salicylate emerge as the three most volatile substances contributing to the aroma of Chimonanthus flowers, accounting for 48%, 26%, and 11% of the total volatile compounds, respectively, corroborating findings from previous studies [26].

### 3.2. Aroma Quality of Three Tea Dhools

To conduct a thorough investigation into the aromatic volatiles present in LLYT, DHJZ, and HSMF dhool, the volatile compounds of these three tea dhools were analyzed using GC-MS. A total of 70, 52, and 57 volatiles were identified in LLYT, DHJZ, and HSMF dhool, respectively. In the LLYT dhool sample, 8 alcohols, 11 aldehydes, 11 alkanes, 4 esters, 11 ketones, 20 nitrogen heterocycles, 4 oxygen heterocycles, and 1 phenolic compound were detected (Appendix A). The DHJZ dhool sample contained 15 alcohols, 15 aldehydes, 7 alkanes, 6 esters, 6 ketones, 1 oxygen heterocycles, and 2 phenolic compounds (Appendix A). Notably, in DHJZ, linalool, linalool oxide I, linalool oxide II, phenethanol, geraniol, phenylethanal, *β*-myrcene, and (E)-*β*-ionone were identified as integral to the aroma profile of high-grade Dianhong tea [25]. Among these key aromatic components, linalool, geraniol, and phenylethanal were highlighted as essential aroma-active compounds contributing to the characteristic floral and caramel-like odors of high-grade Dianhong tea [25]. The HSMF dhool sample revealed the presence of 14 alcohols, 14 aldehydes, 15 alkanes, 3 esters, 9 ketones, 1 nitrogen heterocycles, and 1 oxygen heterocycles compound (Appendix A). The volatile compounds linalool, heptanal, nonanal, decanal, and (E)-*β*-ionone were detected in HSMF dhool, and these five volatiles are regarded as the primary contributors to its aromatic profile [27].

The radar chart illustrates the aromatic characteristics of the three tea dhools (Figure 2A). Analysis of the radar chart indicates that alcohols are the predominant contributors to the aroma quality of DHJZ and HSMF dhool, whereas nitrogen heterocyclic compounds significantly enhance the aroma quality of LLYT dhool. The high proportion of nitrogen heterocyclic compounds serves as a distinctive feature that sets LLYT apart from other teas, and it plays a crucial role in forming the characteristic rice crust flavor. In the LLYT dhool samples, a considerable proportion of nitrogen heterocyclic compounds were detected, including 2,5-dimethyl-pyrazine, 2-ethyl-3-methylpyrazine, 2-ethyl-5-methylpyrazine, 3-ethyl-2,5-dimethylpyrazine, 2,6-diethylpyrazine, 2,5-diethylpyrazine, 2,3-diethyl-5-methylpyrazine, 2-methyl-3,5-diethylpyrazine, and 1-furfurylpyrrole, aligning with findings from prior research [3].

To further elucidate the distinctions in aroma volatiles among LLYT, DHJZ, and HSMF dhools, we performed multivariate statistical analysis. All volatile components were examined utilizing unsupervised PCA (Figure 2B) and HCA (Figure 2C). The PCA score plots exhibited significant differences in volatile components among the three tea dhools, while the HCA dendrogram illustrated that each tea dhool possesses distinct characteristics, and the experimental manipulations demonstrated commendable reproducibility.

### 3.3. Scanning Electron Microscopy (SEM) Examination of Solid Surfaces of Different Tea Dhools

To understand why these three tea dhools had such drastically different adsorption capacity towards the volatiles from Chimonanthus flowers, we examined the surface texture and structures of three tea dhools, and found the surface of LLYT exhibits a wrinkled structural morphology, characterized by irregular grooves. The surface structure of HSMF presents a stacked striated pattern, and the surface pores are notably smaller and narrower compared to LLYT. The surface of DHJZ is covered with a dense array of cuticular trichomes, and a lot of loose pores are formed between the trichomes (Figure 2D).

### 3.4. Contribution of Chimonanthus Flowers Volatiles to Aroma Quality of Three Scented Chimonanthus Teas

The surface of a tea dhool typically exhibits greater wrinkling and a larger surface area, endowing it with favorable adsorption characteristics. The process of scenting tea with flowers in a specific proportion encourages the tea dhools to absorb the floral fragrance, a technique well-known as the scenting process. This method serves as an effective measure to enhance aroma quality and diversify the aromatic profiles of tea. To ascertain the potential influence of Chimonanthus flowers on the aroma quality of three distinct tea dhools, Chimonanthus flowers were meticulously scented with LLYT, DHJZ, and HSMF dhools, respectively, followed by the analysis of volatile compounds in each tea dhool post-scenting using GC-MS (Figure 3). Following the scenting process with Chimonanthus flowers, the aroma composition of tea dhool underwent considerable alteration, with the contributions of alcohol and ester compounds becoming particularly prominent (Appendix A). This phenomenon may be attributed to the adsorption of these compounds contained in the Chimonanthus flowers by the tea dhool.

Given that the three tea dhools exhibit markedly different surface structures and aroma compositions, it is plausible that they possess varying adsorption characteristics. To investigate the adsorption preferences of the three tea dhools to the aromas of Chimonanthus flowers, the accumulation patterns of common aromatic components before and after scenting were analyzed using volcanic diagrams (Figure 4A–C). Specifically, LLYT, DHJZ, and HSMF dhools displayed 40, 31, and 38 common aromatic components, respectively, both before and after scenting with Chimonanthus flowers. The accumulation patterns of these common aromatic components were further analyzed through a heatmap, as illustrated in Figure 4D–F. In post-scenting with Chimonanthus flowers, the content percentage of linalool, methyl salicylate, and indole in the volatiles of LLYT dhool, as well as phenylmethanol, phenylmethyl acetate, methyl salicylate, and eugenol in the volatiles of DHJZ dhool, and phenylmethanol, phenylmethyl acetate, and methyl salicylate in the volatiles of HSMF dhool, exhibited a significant increase (FC > 2, *p* < 0.01). The linalool, methyl salicylate, indole, phenylmethanol, phenylmethyl acetate, and eugenol in the flowers are absorbed by the tea dhools, substantially amplified the role of these volatiles in enhancing aroma quality. Among these, linalool, indole, and phenylmethanol contribute predominantly floral/sweet flavor to Chimonanthus tea, while methyl salicylate and eugenol impart spicy/herbal aromas, and component phenylmethyl acetate provide fruitier flavor (Appendix A). Following scenting, the content percentage of phenylmethyl acetate in DHJZ dhool increased approximately sevenfold, while in HSMF dhool it rose around sixfold. Additionally, the concentration of eugenol in DHJZ dhool also surged by nearly sevenfold.

Subsequently, OPLS-DA was conducted to assess whether the aroma volatiles of the tea samples varied before and after the scenting with Chimonanthus flowers. The outcomes of the analysis indicated that the aroma volatiles of LLYT, DHJZ, and HSMF dhools were significantly different from those of the scented tea infused with Chimonanthus flowers, demonstrating that the volatiles from Chimonanthus flowers effectively transformed the aroma composition of tea dhools (Appendix A). The VIP analysis discerned critical volatile compounds that exhibited significant concentration differences across the samples. Specifically, 17, 14, and 16 distinct volatiles with VIP > 1 were identified in the LLYT, DHJZ, and HSMF samples, respectively, both before and after scenting. This indicates that the scenting of Chimonanthus flowers most profoundly influenced the aroma composition of LLYT, followed by HSMF and DHJZ. To mitigate the risk of overfitting, a 200-bootstrap permutation test was performed, yielding results that confirmed the quality parameters of the generated OPLS-DA model, which displayed a commendable fit with high predictive capability (Appendix A). HCA further illustrated that the tea samples before and after scenting demonstrated distinct classification characteristics (Appendix A).

### 3.5. Identification of New Aroma Components After Scenting of Tea Dhool with Chimonanthus Flowers

Through the analysis of volatile compounds in the three tea dhools before and after the scenting process, it was revealed that 17, 12, and 12 aroma compounds were newly introduced following the scenting of LLYT, DHJZ, and HSMF dhools with Chimonanthus flowers, respectively. Among these, seven newly added compounds were common to all three tea dhools: pulegone, 3-phenylpropanol, (E)-cinnamaldehyde, cinnamyl alcohol, γ-phenylpropyl acetate, (E)-isoeugenol, and (E)-cinnamyl acetate (Appendix A). These findings indicate that the scenting with Chimonanthus flowers imparted a greater array of volatiles to LLYT dhool, thereby enhancing its aroma quality.

### 3.6. Changes of Seven Types of Volatile Substances Added After Scenting During Storage

The seven newly introduced volatiles in LLYT, DHJZ, and HSMF following scenting with Chimonanthus flowers may significantly enhance the aroma quality of the tea; however, the ability of these adsorbed volatiles to remain during storage is crucial in determining whether they confer novel aroma types to the tea. Consequently, we examined the retention of these seven compounds of Chimonanthus tea during storage (Figure 5A–C). Results indicated that the contents of total aroma compounds in Chimonanthus tea decreased, but the loss rates were different for those volatile compounds in all three types of Chimonanthus tea. The content percentage of compounds with slower loss rates increased, while the content percentage of compounds with faster loss rates decreased steadily. For example, after 56 days of storage, the content percentage of pulegone in all three categories of Chimonanthus teas was commendably preserved and increased. Notably, in the scented LLYT-Chimonanthus tea, the content percentage of 3-phenylpropanol, cinnamyl alcohol, and (E)-cinnamyl acetate exhibited the most marked decrease after 28 days, while in the scented DHJZ- and HSMF-Chimonanthus teas, this decline was observed after 42 days. Among the three types of Chimonanthus teas, (E)-cinnamaldehyde displayed the most pronounced reduction after 42 days, and the percentage of γ-phenylpropyl acetate continued its significant decline after 28 days. In the scented LLYT-Chimonanthus tea, (E)-isoeugenol saw a substantial decrease after 42 days of storage, while such a decrease was not significant in the scented DHJZ- and HSMF-Chimonanthus tea.

During the storage of scented LYYT-Chimonanthus tea, the relative contents of 3-phenylpropanol, (E)-cinnamaldehyde, cinnamyl alcohol, *γ*-phenylpropyl acetate, and (E)-cinnamyl acetate decreased sharply after 28 days, diminishing approximately 5-fold, 4-fold, 11-fold, 5-fold, and 8-fold, respectively. However, pulegone and (E)-isoeugenol commenced a pronounced decline after 56 days, diminishing approximately three-fold and six-fold, respectively (Appendix A). In the scented DHJZ, the relative contents of cinnamyl alcohol and (E)-cinnamyl acetate declined sharply after 42 days, diminishing roughly 3-fold and 4-fold, respectively, whereas 3-phenylpropanol, (E)-cinnamaldehyde, *γ*-phenylpropyl acetate, and (E)-isoeugenol began to decrease markedly after 56 days, by approximately 19-fold, 52-fold, 11-fold and 3-fold, respectively (Appendix A). For the scented HSMF-Chimonanthus tea, the relative contents of *γ*-phenylpropyl acetate and (E)-cinnamyl acetate exhibited a sharp decrease after 28 days, decreasing by approximately five-fold and four-fold, respectively; concurrently, 3-phenylpropanol, (E)-cinnamaldehyde, cinnamyl alcohol, and (E)-isoeugenol began to decline markedly after 42 days, by about eight-fold, six-fold, nine-fold, and six-fold, respectively (Appendix A).

These findings suggest that the adsorption capacity of DHJZ dhool surpasses that of HSMF dhool, followed by LYYT dhool. This phenomenon may be attributable to the rough and porous surface structures with extensive coverage of tea trichomes on DHJZ dhools. These surface structures and trichomes facilitate both physical and chemical interactions between floral volatile molecules and tea surface molecules, mainly involving partly damaged cell wall components and cuticular waxes during tea processing [28]. The higher the adsorption capacity of tea dhools for volatile substances, the better the transferring effects of floral volatiles to tea dhools. Furthermore, during storage, the relatively smaller surface pore size but higher pore density on HSMF dhools could be able to constrain more aroma molecules, as compared with the looser, and larger pore size on the LLYT dhool surfaces [29].

### 3.7. Dynamic Changes in Volatiles of Three Chimonanthus Teas During Storage

Different volatiles also exhibited varying accumulation patterns in scented Chimonanthus tea. Using a heatmap, we visualized these accumulation patterns of various volatiles in three scented Chimonanthus teas during storage (Appendix A). In the scented LLYT-Chimonanthus tea samples, the content percentage of pentanol and linalool consistently increased during 56 days of storage, while the content percentage of 3-phenylpropanol, cinnamyl alcohol, phenylmethyl acetate, *γ*-phenylpropyl acetate, (E)-cinnamyl acetate, and 2,3-dihydrobenzofuran continuously declined (Appendix A). In the scented DHJZ-Chimonanthus tea samples, the content percentage of 3-octenol, linalool oxide I, linalool oxide II, linalool, linalool oxide IV, 4-terpineol, 3-methyl-butanal, phenylethanal, styrene, *β*-myrcene, pulegone, and 4-amino-2,6-dimethylphenol continued to rise throughout the 56 days of storage, whereas (E)-cinnamaldehyde, phenylmethyl acetate, (E)-cinnamyl acetate, and indole consistently diminished (Appendix A). In the scented HSMF-Chimonanthus tea samples, the content percentage of linalool steadily increased over the storage, while the content percentage of R-limonene, phenylmethyl acetate, and *α*-ionone showed a continuous decrease (Appendix A). Among the three scented Chimonanthus teas, the content percentage of methyl salicylate showed only a slight increase after 14 days of storage, followed by a significant decline during the subsequent storage period. It is worth noting that the increase in the content percentage of some compounds, such as pentanol, linalool, and linalool oxides, during storage may be due to their significantly lower loss rate compared to other compounds.

### 3.8. Changes in Different Volatile Compounds in Three Chimonanthus Teas During Storage After Scenting

A total of eight types of volatile compounds were identified in the tea samples and tea samples stored at different times after scenting, and their corresponding content percentages are illustrated in Figure 6A–C. A comparison of the content percentage of different volatile substances within the three tea dhools prior to scenting revealed that nitrogen heterocyclic compounds constituted the most significant proportion in the LLYT-Chimonanthus tea, at approximately 52%. This particular composition contributes to the rice crust and roasted flavor of LLYT-Chimonanthus tea, playing a crucial role in shaping its aroma quality (Appendix A, Figure 6D). In contrast, alcohol compounds were the predominant volatile type in DHJZ-Chimonanthus tea (approximately 72%) and HSMF-Chimonanthus tea (approximately 45%), serving as essential contributors to their respective floral/sweet aromas (Appendix A).

Interestingly, the aromatic volatiles from Chimonanthus flowers were absorbed by the tea dhools, significantly altering the aromatic composition of the tea dhools. This transformation was primarily attributed to the increase in ester compounds, which enriched a greater floral and fruity flavor to the three varieties of tea dhool (Figure 6D, Appendix A). After scenting, the content percentage of esters in LLYT, DHJZ, and HSMF increased from 7% to 60%, from 10% to 57%, and from 5% to 56%, respectively (Figure 6A–C). Phenylmethyl acetate and methyl salicylate, present in the Chimonanthus flowers, were the primary volatiles responsible for the significant increases in ester compounds in Chimonanthus teas after scenting. These two compounds were also the most prevalent in the total volatiles of the Chimonanthus flowers, accounting for approximately 26% and 9%, respectively (Appendix A). However, the aroma composition of Chimonanthus teas underwent substantial changes during subsequent storage. In the LLYT-, DHJZ-, and HSMF-Chimonanthus teas after scenting and during storage, the content percentage of ester compounds decreased continuously during storage, reduced by approximately 56-fold, 11-fold, and 30-fold, respectively. This result indicates that the abundant trichomes covering the DHJZ dhool surfaces facilitated not only adsorption, but also constrained these adsorbed ester aroma compounds in Chimonanthus teas, whereas the loose and large porosity in the dhool surfaces of LLYT favored a rapid adsorption of ester aroma components, albeit with a diminished retention capacity during Chimonanthus tea storage. The smaller pore structure on the surface of HSMF exhibits adsorption and retention advantages for ester aromas, but the overall effects lie between those of LLYT- and DHJZ-Chimonanthus teas. Unlike ester volatiles, the content percentage of alcohol compounds in LLYT-, DHJZ- and HSMF-Chimonanthus teas increased by approximately 3.2-fold, 2.7-fold, and 2.6-fold, respectively, during storage. After 56 days of storage, the alcohol compounds became the dominant compounds, which may impart a more floral and sweeter flavor to the tea [25]. Compared to the tea without scenting, the content percentage of alcohol compounds in LLYT, DHJZ, and HSMF stored for 56 days increased by approximately six-fold, one-fold, and two-fold, respectively. These results indicate that esters and alcohols, which comprise the primary aroma types of Chimonanthus flowers, are the two types of volatiles that influence the aroma composition of the tea.

Unlike ester volatiles, which are challenging to maintain after adsorption, alcohol compounds can enhance the aroma quality of the tea for a longer period after adsorption, and the improvement effect on the aroma quality of LLYT is significantly higher than that of HSMF and DHJZ. Therefore, alcohols and esters in the Chimonanthus flowers volatiles are the key compounds that alter the aroma composition of the three types of tea, with esters modifying the aroma quality of the Chimonanthus tea in the early stage of storage, while alcohols are the primary contributors to improving the original aroma of the tea in the later stages of storage.

## 4. Discussion and Conclusions

The sweet and aromatic *Chimonanthus praecox* flower is famous for its multifaceted applications in ornamental, medicinal, cosmetic, and culinary realms, attributed to the presence of various bioactive volatiles and alkaloids, such as calycanthine, found in its flowers, leaves, and roots [30,31]. It is traditionally employed in the treatment of coughs and rheumatic arthritis, showcasing bioactivities including antioxidative, anticarcinogenic, and antihypertensive properties within the framework of traditional Chinese medicine [11]. Consequently, the innovative Chimonanthus teas have significantly enriched their value by offering additional fragrance and health-promoting bioactivity.

Adsorption of flower fragrance molecules by tea dhools varies with different surface matrix properties. Generally, the interaction between gas molecules and solid surfaces is of great significance in the flower-scenting of tea dhool. The interaction models not only involve the volatile molecule adherence to and attachment with tea dhool surfaces, or penetration through tea dhool trichome layers and trapped into the porous structures in their surfaces [28,29]. The models also include interactions between flower volatile molecules and aroma molecules emitted from tea dhools themselves, enrichment or expulsion. When flower volatile molecules contact the tea dhool surface, some molecules will be adsorbed on the surface, via physical adsorption occurring through van der Waals forces with weaker adsorption forces, or chemical adsorption, occurring through forming chemical bonds with stronger adsorption forces. The interactions between flower fragrance molecules and tea dhool surfaces may also occur through electric charge and elastic interactions. The electric charges cause some volatile molecules to be adsorbed on tea dhool surfaces, while the elastic interaction affects the rebound and penetration rate of volatile molecules through tea dhool surface structures. Among the main factors affecting the interaction between flower molecules and solid surfaces of tea dhools, the surface smoothness or roughness, porous structures (size and density), and the nature of the charge carried are of great matter [29]. The rough surfaces of tea dhools provide more adsorption sites, increasing adsorption and holding capacity of volatile molecules, directly affecting the flower scenting effects. By contrast, the smooth surfaces typically result in fewer adsorption sites and lower adsorption capacity. The porous structure formed by processing damages or trichomes on tea dhool surfaces have a significant impact on the adsorption of flower molecules. For example, HSMF formed small and narrow pores on its surface after rolling, resulting in a slower adsorption of floral aroma molecules, yet exhibiting good retention during storage. In contrast, LLYT, which is not rolling, presents larger and looser pores, allowing for rapid adsorption of floral aroma molecules, but its ability to retain aromatic molecules is poor. Moreover, DHJZ, although not rolling, possesses a densely covered surface with trichomes that effectively prevent the dissipation of floral aroma molecules. Porous materials of tea dhools can provide more adsorption capacity [32]. The size and distribution of pores also affect adsorption behavior, with micropores, mesopores, and macropores having different adsorption capacities for different volatile molecules [29]. The nature of the charge carried by tea dhool surfaces can affect the adsorption of volatile molecules. A surface with a positive charge will attract volatile molecules with a negative charge and vice versa [33]. This charge interaction further affects the adsorption behavior of flower volatile molecules on tea dhool surfaces.

The abundant trichomes covering the DHJZ dhool surfaces facilitated not only adsorption, but also constrained these adsorbed ester aroma compounds in Chimonanthus teas, whereas the loose and large porosity in the dhool surfaces of LLYT favored a rapid adsorption of ester aroma components, albeit with a diminished retention capacity during LLYT-Chimonanthus tea storage. The smaller pore structures on the surface of HSMF exhibit an adsorption and retention advantages for ester aromas, but the overall effects lie between those of LLYT- and DHJZ-Chimonanthus teas. In order to improve the overall flower scenting performance and aroma-retention in LLYT-Chimonanthus tea to restrict the loss of aroma molecules during storage, improving the shaping process of LLYT by adding a rolling or pressuring procedure to forming tea dhool surface with smaller pores, higher porosity density, and tighter textures, is suggested here.

The scenting of various tea dhools with Chimonanthus flowers enhanced key aroma compounds in the resultant Chimonanthus teas. Notably, the content percentages of linalool, methyl salicylate, and indole in the LLYT-Chimonanthus tea, and phenylmethanol, phenylmethyl acetate, and eugenol in DHJZ-Chimonanthus tea exhibited remarkable increases. The contents of phenylmethanol, phenylmethyl acetate, and methyl salicylate also rose significantly in HSMF-Chimonanthus tea. A comparative analysis of volatiles from the three tea dhools before and after scenting revealed that volatile molecule species and quantities of volatiles were absorbed differentially from Chimonanthus flowers into LLYT, DHJZ, and HSMF dhools. During the dynamic absorption, seven compounds were identified as common molecules absorbed into three tea dhools: pulegone, 3-phenylpropanol, (E)-cinnamaldehyde, cinnamyl alcohol, *γ*-phenylpropyl acetate, (E)-isoeugenol, and (E)-cinnamyl acetate, all present in these scented Chimonanthus teas. These data suggest that physical textures or physiochemical properties of the three tea dhools exhibit largely similar, in terms of their absorption preference. However, other volatiles that are differentially enriched in three Chimonanthus teas indicated their varying absorption preferences and compatibility with the volatiles present in Chimonanthus flowers.

During the storage of Chimonanthus tea, the desorption, trapping/adhesion, release and loss dynamics of aroma volatiles could determine the retention patterns of Chimonanthus tea aroma. The retention patterns of seven major aromatic components shared across the three types of Chimonanthus tea varied distinctively. Specifically, the content percentage of (E)-cinnamaldehyde, *γ*-phenylpropyl acetate, and (E)-isoeugenol exhibited differential decreases among the three Chimonanthus tea types, with the most significant decrease observed in LLYT, indicating that the adsorption capacity and retention of DHJZ and HSMF were superior to that of LLYT. Thus, it can be inferred that DHJZ and HSMF may serve as more suitable tea dhools for the production of Chimonanthus tea.

The variations in absorption patterns and compatibility of different volatiles from Chimonanthus flowers with the three tea dhools subsequently lead to diverse accumulation patterns within the Chimonanthus teas. These aromatic components predominantly included several ester compounds that saw significant increases. Phenylmethyl acetate and methyl salicylate emerged as the principal volatiles, which account for the highest proportions within the total volatiles of Chimonanthus flowers, and were readily absorbed by all tea types, contributing to the Chimonanthus flower aroma in the scented teas.

Similarly to other teas, the aroma of Chimonanthus teas diminished over the course of storage due to the evaporation of volatiles and other factors. In scented LLYT-, DHJZ-, and HSMF-Chimonanthus teas, the content percentage of ester compounds decreased substantially during storage, reaching approximately 56-fold, 11-fold, and 30-fold lower by day 56, respectively. In contrast, the proportion of alcohol compounds in the scented LLYT, DHJZ, and HSMF increased by approximately threefold at day 56, thereby positioning alcohol compounds as the predominant volatile molecules after this duration. This shift may impart a more floral and sweet flavor profile to the tea [25]. Comparatively, the volatiles in the scented LLYT, DHJZ, and HSMF stored for 56 days exhibited increases by about six-fold, one-fold, and two-fold, respectively, compared to non-scented tea. Overall, our findings indicate that esters and alcohols, which constitute the primary aroma types of Chimonanthus flowers, are two principal categories of volatiles that are significantly absorbed by tea dhool and contribute to the aromatic composition of Chimonanthus tea. Unlike the ester volatiles, which appear challenging to retain in Chimonanthus teas following adsorption and prolonged storage, alcohol compounds enhance the aroma quality of tea dhool for an extended period post-adsorption.

By comprehensively comparing volatile molecules in Chimonanthus flowers and different tea dhools before, during, and after Chimonanthus flower scenting process, as well as 56-day storage of Chimonanthus teas, we not only revealed that floral aroma molecules were differentially absorbed by three tea dhools, and characterized the typical aroma components in Chimonanthus flowers and Chimonanthus teas, we also traced the dynamic changes in volatile components during the scenting process and aroma retention in Chimonanthus teas during storage. Furthermore, this study explored the effects of different surface structures/properties of tea dhools, such as pores, roughness or smoothness, as displayed with scanning electron microscopy, on the adsorption and adhesion capacity for floral volatile molecules. The aroma-retention ability of Chimonanthus teas to constrain the loss of aroma molecules during storage is also closely related to physical and chemical properties of tea dhool surfaces. Particularly to LLYT, according to our results about the surface structure–aroma adsorption/retention efficiency relationship, this study provides practical methods to improve aroma quality and health function of LLYT via efficient scenting with Chimonanthus flowers. Furthermore, to improve adsorption capacity and aroma retention/adhesion ability of LLYT-Chimonanthus tea processing, changing LLYT’s surface structures, by improving LLYT shaping process to form smaller pores, higher porosity density, and tighter textures, which could enhance floral aroma adsorption, and aroma retention/adhesion to reduce the loss of volatiles during Chimonanthus tea storage.

In conclusion, we created different types of Chimonanthus tea by scenting three varieties of tea dhools with freshly blooming Chimonanthus flowers. The scenting of tea dhools with Chimonanthus flowers notably enhanced the aroma spectrum, sensory qualities, and longevity of Chimonanthus teas, and likely intensified their potential health benefits. The study augments several critical aromatic components in Chimonanthus teas in comparison to their corresponding non-scented tea dhools, paving a road toward the improvement of LLYT aroma and health benefits and consumption popularity.

## Figures and Tables

**Figure 1 foods-14-01696-f001:**
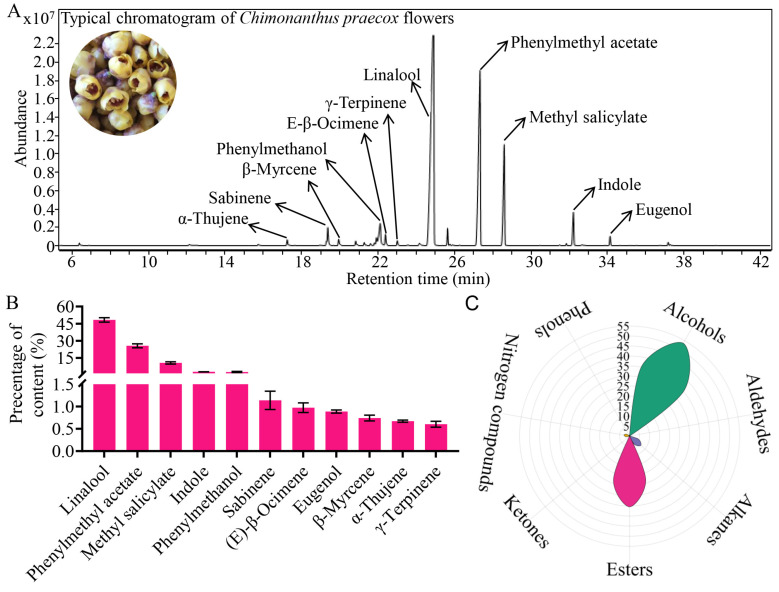
Identification of characteristic aroma of *Chimonanthus praecox* flowers. (**A**) GC-MS traces of aroma in *Chimonanthus praecox* flowers. (**B**) Content percentage of main aroma substances in *C. praecox* flowers. (C) Categories and distribution of volatile compounds. Data were from three experiments (*n*=3) and expressed as means ± s.d.

**Figure 2 foods-14-01696-f002:**
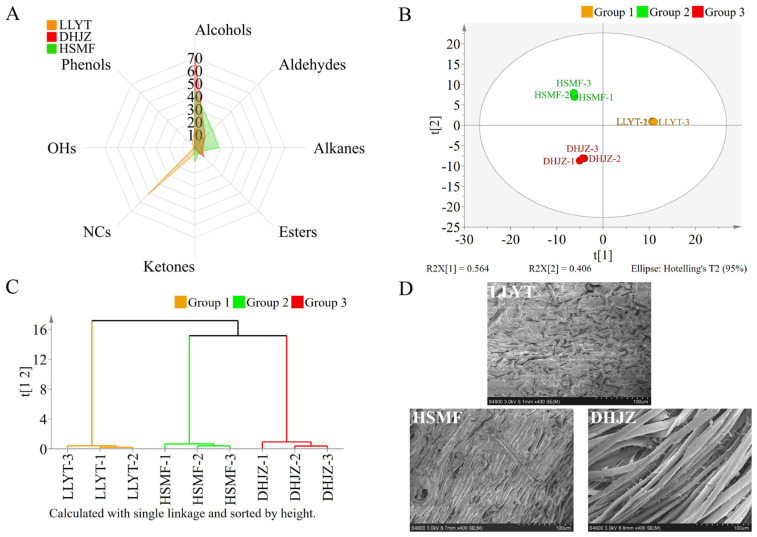
Difference analysis of aroma quality of LLYT, DHJZ, and HSMF dhools. (**A**) Radar chart of distribution of different compounds in LLYT, DHJZ, and HSMF dhools. (**B**) A principal component analysis (PCA) score plot of volatile quality components of LLYT, DHJZ, and HSMF dhools. (**C**) Hierarchical cluster analysis (HCA) of volatile quality components of LLYT, DHJZ, and HSMF dhools. (**D**) The surface structure of LLYT, DHJZ, and HSMF dhools observed under a scanning electron microscopy. Abbreviations: LLYT, Large−leaf Yellow Tea; DHJZ, Dianhong jinzhen Tea; HSMF, Huangshan Maofeng Tea; NCs, Nitrogen compounds; OHs, Oxygen heterocyclics.

**Figure 3 foods-14-01696-f003:**
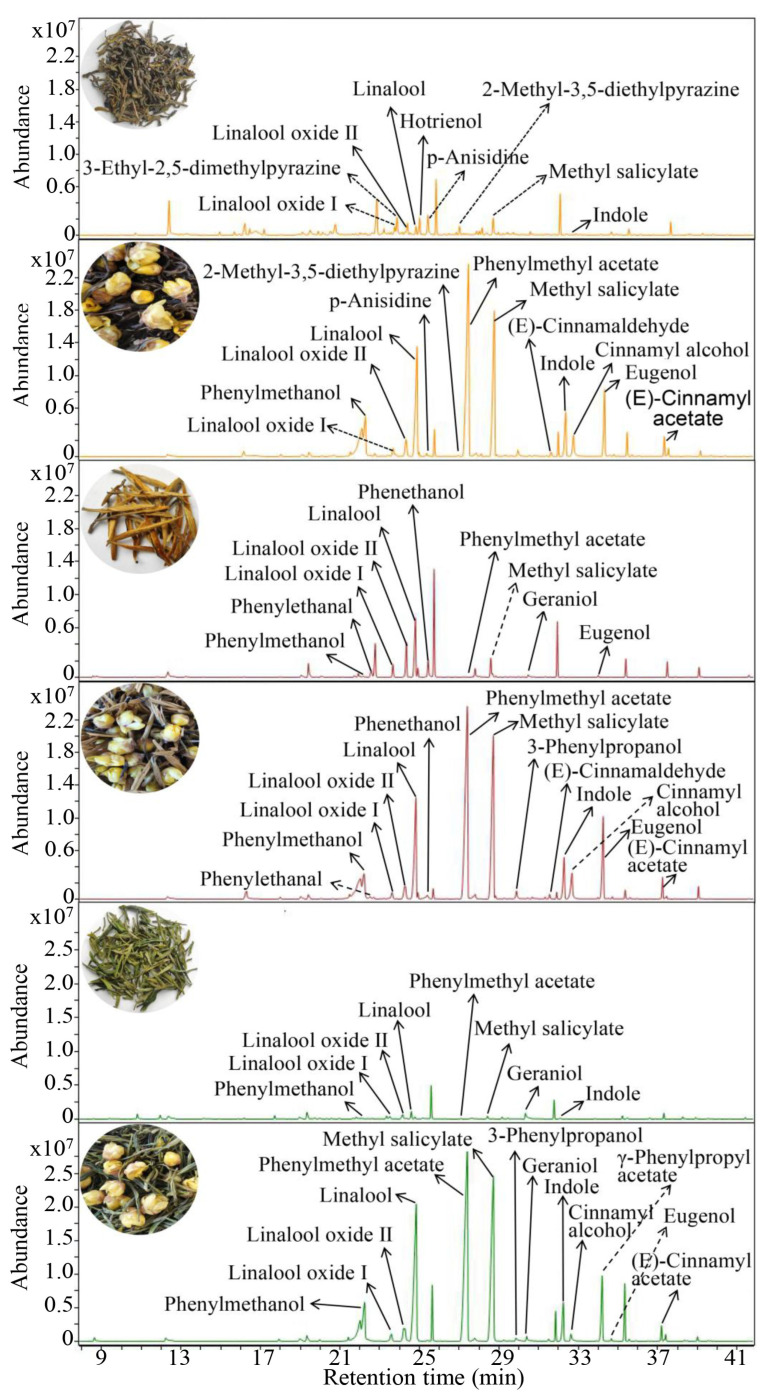
GC-MS traces of aroma components in three tea dhools before and after scenting with Chimonanthus flowers.

**Figure 4 foods-14-01696-f004:**
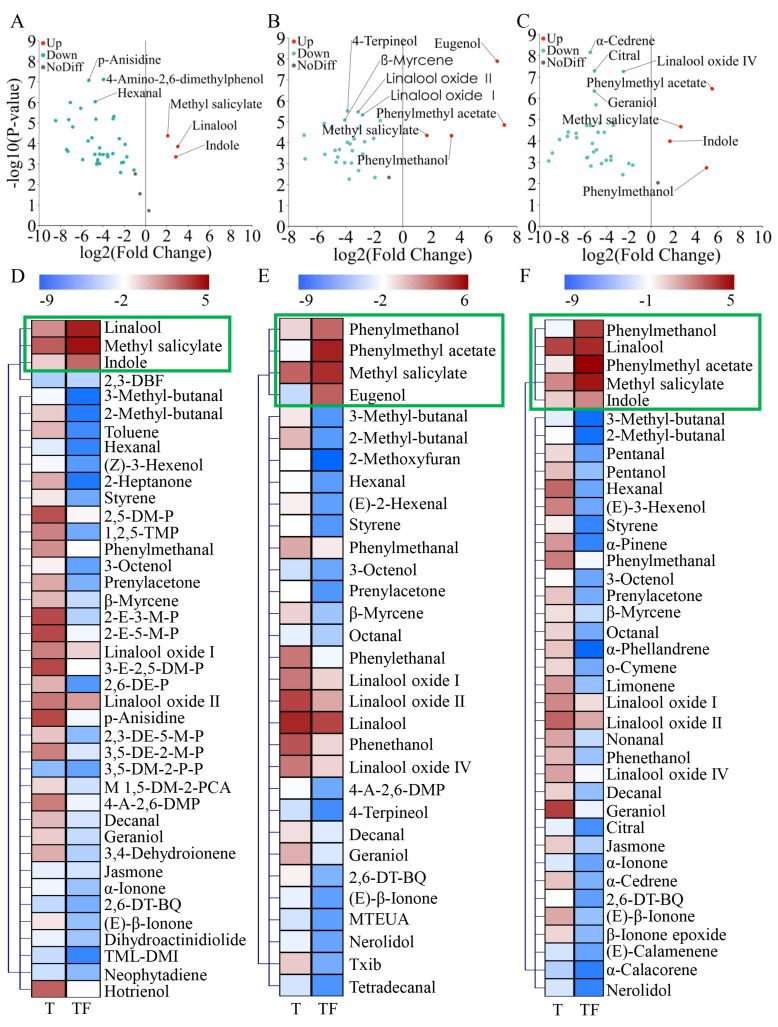
Difference analysis of common aroma components of LLYT, DHJZ, and HSMF before and after scenting with Chimonanthus flowers. (**A**) Volcanic diagram of different compounds in LLYT. (**B**) Volcanic diagram of different compounds in DHJZ. (**C**) Volcanic diagram of different compounds in HSMF. (**D**) Accumulation pattern of common volatiles in LLYT before and after scenting. (**E**) Accumulation pattern of common volatiles in DHJZ before and after scenting. (**F**) Accumulation pattern of common volatiles in HSMF before and after scenting. Abbreviations: T stands for tea without scenting; TF stands for scented tea; 2,3−DBF/2,3−Dihydrobenzofuran; 2,5−DM−P/2,5−Dimethylpyrazine; 1,2,5−TMP/1,2,5−Trimethylpyrrole; 2−E−3−M−P/2−Ethyl−3−methylpyrazine; 2−E−5−M−P/2−Ethyl−5−methylpyrazine; 3−E−2,5−DM−P/3−Ethyl−2,5−dimethylpyrazine; 2,6−DE−P/2,6−Diethylpyrazine; 2,3−DE−5−M−P/2,3−Diethyl−5−methylpyrazine; 3,5−DE−2−M−P/2−Methyl−3,5−diethylpyrazine; 3,5−DM−2−P−P/3,5−Dimethyl−2−propylpyrazine; M 1,5−DM−2−PCA/Methyl 1,5−dimethyl−2−pyrrolecarboxylate; 4−A−2,6−DMP/4−Amino−2,6−dimethylphenol; 2,6−DT−BQ/2,6−Di−tert−butylquinone; TML−DMI/1,7−Trimethylene−2,3−dimethylindole; MTEUA/Methyl 2,4,6−trimethylundecanoate.

**Figure 5 foods-14-01696-f005:**
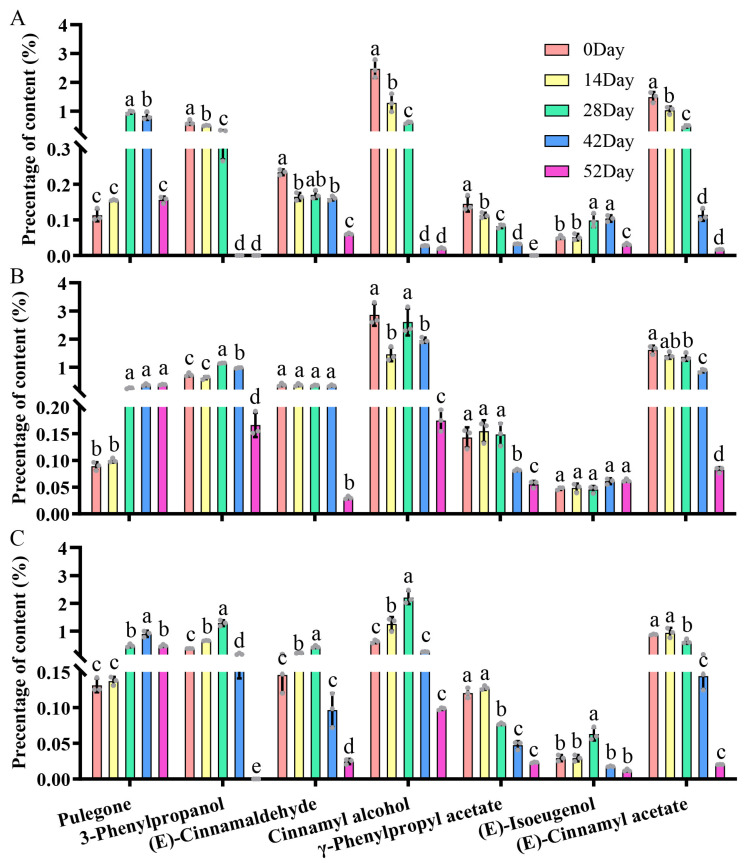
Changes of seven volatile molecules absorbed into Chimonanthus flowers-scented tea during storage. (**A**) Dynamic changes in content percentage of seven volatile molecules in scented LLYT during storage. (**B**) Dynamic changes in content percentage of seven volatile molecules in scented DHJZ during storage. (**C**) Dynamic changes in content percentage of seven volatile molecules in scented HSMF during storage. Data were from three experiments (*n*=3) and expressed as means ± s.d., and the significant differences are indicated by letters from one-way ANOVA followed by Tukey’s HSD test; the different letters of the same compound indicate significant difference at *p* < 0.05.

**Figure 6 foods-14-01696-f006:**
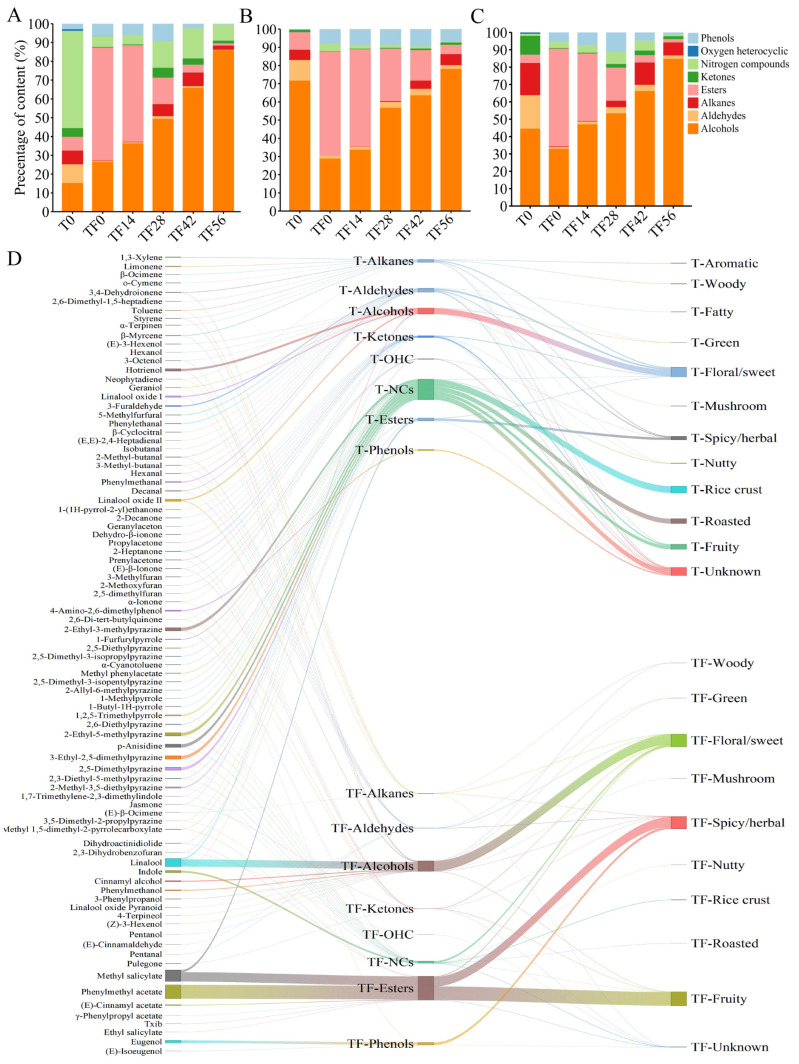
Changes in different volatile compounds in Chimonanthus flowers-scented LLYT, DHJZ and HSMF during storage. (**A**) Changes in content percentage of eight volatile compounds in scented LLYT during storage. **(B**) Changes in content percentage of eight volatile compounds in scented DHJZ during storage. (**C**) Changes in content percentage of eight volatile compounds in scented HSMF during storage. (**D**) LLYT dhool vs. scented LLYT flavor sankey diagram. The index of flavor sankey diagram is as follows: the bars on the left represent volatile substances that contribute to the flavor; the right represents the flavor characteristics; the middle bar represents the categories of aroma volatiles in tea and scented tea; the width of a line indicates the content percentage of the corresponding volatile substance. Abbreviations: T stands for tea without scenting; TF stands for scented tea; 0, 14, 28, 42, and 56 represent the number of days of storage, NCs, Nitrogen compounds, OHC, Oxygen heterocyclic.

## Data Availability

The most data that support the findings of this study are available in the Appendix A of this article, others will be available upon request from the corresponding author, who will also be responsible for distribution of materials integral to the findings presented in this article in accordance with the journal policy described in the Instructions for Authors: Jian Zhao (jzhao2@qq.com; zhaojian@hunau.edu.cn).

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
