# Peer review of "Dynamic Change of Aroma Components in Chimonanthus praecox Flower Scented Teas During Absorption and Storage"

_foods, 2025, doi:10.3390/foods14101696_

Round 1
Reviewer 1 Report
Comments and Suggestions for Authors
While the study on Chimonanthus praecox and its use in flower-scented teas is valuable, there is significant overlap with an earlier publication by the same authors, which raises concerns regarding self-plagiarism. Nonetheless, after reviewing the paper, I believe it requires major revision to clearly differentiate the current study from the previous work (Novel Chimonanthus teas made from scenting different tea dhools with Chimonanthus praecox flowers, published in Food Chemistry. https://www.sciencedirect.com/science/article/pii/S030881462501369X)
Below are suggestions for improving various sections to clarify the novelty of this study and to address issues of repetition.
Introduction
- The introduction provides important background information on Chimonanthus praecox and its role in flower-scented teas, an area that has attracted growing research interest. However, there is notable overlap with the authors' earlier publication, particularly in describing the health benefits of Chimonanthus flowers, their phytochemical composition, and their use in tea. While it is understandable that the authors may want to reference their own previous work, it is essential to either paraphrase these sections more thoroughly or acknowledge the earlier work explicitly to avoid the perception of self-plagiarism.
- The flow between the sections could be improved. Moreover, it would be very suitable a clearer statement of the research gap that this study addresses.
- I would recommend that the authors re-examine this section. While there is no explicit duplication of content, there is significant overlap in phrasing and ideas with their previous work
Materials and Methods
- Again I have noticed a substantial overlap with your previously published work. Many sections of the methodology appear to be almost identical. While it is common to build upon prior research, the extensive reuse of these details, without clear indication of prior publication, raises concerns about self-plagiarism.
- The authors should cite the previous work and clearly indicate how the methodology in this paper either adapts or replicates the previous research.
- More detailed explanations or novel insights in the methodology could help distinguish this work from the earlier publication.Results
Results
Section 3.1. This section shows significant similarities with the volatile component analysis presented in the earlier work. Both studies identify a similar set of volatile compounds, such as linalool, methyl salicylate, and phenylmethyl acetate, as key contributors to the aroma of Chimonanthus flowers. While consistency in results is valuable, the repetition of methods and findings from the earlier paper could be problematic. The authors should highlight how the current study builds on or extends these findings and clearly differentiate the two studies.
Section 3.2. The volatile profiles of the different tea dhools, using GC-MS and multivariate analysis, are valuable contributions. However, the overlap with earlier studies on similar tea types requires further clarification. The findings on LLYT and DHJZ should be explicitly distinguished from prior work to demonstrate the novel aspects of the current research. The authors should also connect the volatile compounds to sensory characteristics, which would enhance the practical relevance of the study.
Section 3.3. OPLS-DA and VIP analyses provide interesting insights into volatile profiles,, a more detailed interpretation of the significance of key volatile compounds identified (e.g., linalool, methyl salicylate and phenylmethanol) should have been added by linking them to their sensory impact in relation to the sensory attributes of the tea.
A deeper discussion on how the surface structure of tea dhools influences volatile adsorption would also be valuable in adding context to the findings. The authors briefly touch on surface structure differences but could expand on how these variations lead to different adsorption patterns between LLYT, DHJZ, and HSMF dhools.
The figures, such as heatmaps and volcanic diagrams, are mentioned but could benefit from a clearer explanation of what specific insights the reader should gain from them.
Section 3.4. The identification of new aroma components is an interesting contribution. However, the authors could provide more insight into how these newly introduced volatiles alter the sensory profile of the tea.
Section 3.5. The authors should clarify the molecular mechanisms behind volatile retention, particularly how structural differences in the tea dhools (e.g., trichomes, pore size) affect this process. The connection between volatile retention and sensory changes needs to be made more explicit.
Section 3.6. This section provides useful insights, but there is significant overlap with the earlier work, particularly regarding the HSMF tea. The authors should emphasize any new findings and avoid repeating data already published in previous studies.
Section 3.7. The discussion on ester and alcohol compounds overlaps with the earlier work. The authors should clarify how these volatiles affect the sensory profile of tea over time, making sure that the current study’s contributions are distinct from the prior research.
Discussion and Conclusions
The discussion and conclusions section provides a comprehensive overview of the findings regarding the volatile profiles of Chimonanthus-scented teas. However, there are concerns about excessive overlap with previous work by the same authors, especially in discussing the volatile compounds in Chimonanthus flowers and their absorption by tea dhools. While the study contributes to the understanding of the interaction between Chimonanthus flowers and tea dhools, some of the results—such as the identification of common aroma compounds—appear to reiterate conclusions from earlier publications without sufficiently distinguishing the novelty of the current research. Additionally, the conclusion regarding the aroma persistence of Chimonanthus teas during storage is valuable, but it would benefit from more detailed comparisons to prior studies to highlight any new insights. To enhance the manuscript, the authors should consider clarifying how the current findings offer new contributions beyond what has already been established in their previous work
Comments on the Quality of English LanguageThe English and writing can be improved, especially the connections between ideas, paragraphs, and sections.
Author Response
Point-by-point responses to reviewers' comments
Reviewers' comments:
Comments and Suggestions for Authors
While the study on Chimonanthus praecox and its use in flower-scented teas is valuable, there is significant overlap with an earlier publication by the same authors, which raises concerns regarding self-plagiarism. Nonetheless, after reviewing the paper, I believe it requires major revision to clearly differentiate the current study from the previous work (Novel Chimonanthus teas made from scenting different tea dhools with Chimonanthus praecox flowers, published in Food Chemistry. https://www.sciencedirect.com/science/article/pii/S030881462501369X)
Below are suggestions for improving various sections to clarify the novelty of this study and to address issues of repetition.
Introduction
The introduction provides important background information on Chimonanthus praecox and its role in flower-scented teas, an area that has attracted growing research interest. However, there is notable overlap with the authors' earlier publication, particularly in describing the health benefits of Chimonanthus flowers, their phytochemical composition, and their use in tea. While it is understandable that the authors may want to reference their own previous work, it is essential to either paraphrase these sections more thoroughly or acknowledge the earlier work explicitly to avoid the perception of self-plagiarism.
The flow between the sections could be improved. Moreover, it would be very suitable a clearer statement of the research gap that this study addresses.
I would recommend that the authors re-examine this section. While there is no explicit duplication of content, there is significant overlap in phrasing and ideas with their previous work
Response: The introduction has been rewritten to emphasize the contribution of Chimonanthus flowers to the enhancement of large-leaf yellow tea quality. Additionally, extraneous descriptions concerning the medicinal value of Chimonanthus flowers have been removed, along with the citation of recent publications.
Materials and Methods
Again I have noticed a substantial overlap with your previously published work. Many sections of the methodology appear to be almost identical. While it is common to build upon prior research, the extensive reuse of these details, without clear indication of prior publication, raises concerns about self-plagiarism.
The authors should cite the previous work and clearly indicate how the methodology in this paper either adapts or replicates the previous research.
More detailed explanations or novel insights in the methodology could help distinguish this work from the earlier publication.
Response: The methods section now includes citations from recently published relevant articles. This section encompasses the processing methods of Chimonanthus tea, the GC-MS detection methods, and the identification techniques for volatile compounds, all derived from recently research, as these methodologies have reached a level of maturity. Notably, the Chimonanthus flowers and tea dhools utilized in this study differ significantly from those employed in previous article. Furthermore, it is important to highlight that this article focuses on the contribution of Chimonanthus flowers to the quality of large-leaf yellow tea and the potential influence of the micro-structure of tea on aroma adsorption, which marks a considerable departure from earlier studies.
Results
Section 3.1.
This section shows significant similarities with the volatile component analysis presented in the earlier work. Both studies identify a similar set of volatile compounds, such as linalool, methyl salicylate, and phenylmethyl acetate, as key contributors to the aroma of Chimonanthus flowers. While consistency in results is valuable, the repetition of methods and findings from the earlier paper could be problematic. The authors should highlight how the current study builds on or extends these findings and clearly differentiate the two studies.
Response: In previous articles, a systematic analysis was conducted to elucidate the differences in the volatile composition between Chimonanthus praecox var. Concolor and Chimonanthus praecox var. Grandiflorus, ultimately focusing on Chimonanthus praecox var. Concolor as the primary subject of study. Building upon prior research, this investigation primarily utilized Chimonanthus praecox var. Grandiflorus for the processing of scented tea. Notably, the harvesting time for Chimonanthus praecox var. Grandiflorus differed from that in earlier studies; therefore, while the main components of its volatiles, such aslinalool, methyl salicylate, and phenylmethyl acetate, and (E)-β-ocimene, remained largely unchanged, other volatiles or those present in lower concentrations exhibited significant variation.
Section 3.2.
The volatile profiles of the different tea dhools, using GC-MS and multivariate analysis, are valuable contributions. However, the overlap with earlier studies on similar tea types requires further clarification. The findings on LLYT and DHJZ should be explicitly distinguished from prior work to demonstrate the novel aspects of the current research. The authors should also connect the volatile compounds to sensory characteristics, which would enhance the practical relevance of the study.
Response: The HSMF utilized in this study differed in purchase timing from that employed in prior research, and exhibited certain discrepancies in aroma composition compared to recently published articles. Both samples share 32 common aromatic components, while the HSMF analyzed in this study possesses 25 unique aromatic constituents. This indicates that the HSMF from the two studies has distinct aromatic profiles, as illustrated in the figure below.
This article aims to elucidate the role of the fragrance of Chimonanthus praecox flowers in enhancing the quality of renowned teas from Anhui Province (HSMF and LLYT). Consequently, HSMF was selected as the primary tea medium, while DHJZ was chosen for its exceptional aroma and the presence of abundant tea hairs, serving as a control to investigate the potential role of tea hairs in the adsorption of Chimonanthus flowers volatiles.
Due to limitations in technical personnel and equipment, a sensory evaluation was not conducted. Instead, a detailed representation of the aroma profile of Chimonanthus tea was illustrated using a flavor Sankey diagram, to elucidate the influence of Chimonanthus praecox flowers volatiles on the aromatic composition of the tea.
Section 3.3.
OPLS-DA and VIP analyses provide interesting insights into volatile profiles,, a more detailed interpretation of the significance of key volatile compounds identified (e.g., linalool, methyl salicylate and phenylmethanol) should have been added by linking them to their sensory impact in relation to the sensory attributes of the tea.
Response: A description of the contribution of key volatile compounds to the aroma composition of Chimonanthus tea has been added in the corresponding section of the article.
A deeper discussion on how the surface structure of tea dhools influences volatile adsorption would also be valuable in adding context to the findings. The authors briefly touch on surface structure differences but could expand on how these variations lead to different adsorption patterns between LLYT, DHJZ, and HSMF dhools.
Response: A description of the contribution of the surface structure of tea dhools to aroma adsorption has been added in the corresponding chapter of the article.
The figures, such as heatmaps and volcanic diagrams, are mentioned but could benefit from a clearer explanation of what specific insights the reader should gain from them.
Response: These figures aim to illustrate certain compounds, such as linalool, methyl salicylate, indole, phenylmethanol, phenylmethyl acetate, and eugenol, that exhibit significant increases after scenting with Chimonanthus flowers. The substantial adsorption of these volatiles has greatly transformed the original aromatic composition of the tea. Those volatiles deserving of the reader's attention are highlighted in the figures using text or green boxes.
Section 3.4.
The identification of new aroma components is an interesting contribution. However, the authors could provide more insight into how these newly introduced volatiles alter the sensory profile of the tea.
Response: The contribution of these new aroma components to the aroma quality of Chimonanthus tea is concentrated in Figure 6D, as well as Figure S8 and Figure S9.
Section 3.5.
The authors should clarify the molecular mechanisms behind volatile retention, particularly how structural differences in the tea dhools (e.g., trichomes, pore size) affect this process. The connection between volatile retention and sensory changes needs to be made more explicit.
Response: The role of the surface structure of tea dhools in the retention of volatiles has been integrated into this chapter, and the potential interactions occurring on the surface of tea dhools have been elaborated upon in the discussion section.
The aromatic volatiles referenced in this chapter, which contribute to the fragrance profile of Chimonanthus tea, are illustrated in Figure 6D, as well as Figure S8 and Figure S9.. The loss of these components would diminish their respective contributions to the sensory quality of Chimonanthus tea.
Section 3.6.
This section provides useful insights, but there is significant overlap with the earlier work, particularly regarding the HSMF tea. The authors should emphasize any new findings and avoid repeating data already published in previous studies.
Response: Please refer to section 3.2 above for responses to similar questions.
Section 3.7.
The discussion on ester and alcohol compounds overlaps with the earlier work. The authors should clarify how these volatiles affect the sensory profile of tea over time, making sure that the current study’s contributions are distinct from the prior research.
Response: Alcohols and esters are the primary classes of compounds constituting the aroma profile of Chimonanthus tea, and their variations during storage are notably pronounced, consistent with our previous research. However, this chapter distinctly emphasizes the influence of volatile compounds from Chimonanthus flowers on the aromatic composition of LLYT, as well as the impact of the surface structure of tea dhools on the adsorption of esters aroma constituents.
Discussion and Conclusions
The discussion and conclusions section provides a comprehensive overview of the findings regarding the volatile profiles of Chimonanthus-scented teas. However, there are concerns about excessive overlap with previous work by the same authors, especially in discussing the volatile compounds in Chimonanthus flowers and their absorption by tea dhools. While the study contributes to the understanding of the interaction between Chimonanthus flowers and tea dhools, some of the results—such as the identification of common aroma compounds—appear to reiterate conclusions from earlier publications without sufficiently distinguishing the novelty of the current research. Additionally, the conclusion regarding the aroma persistence of Chimonanthus teas during storage is valuable, but it would benefit from more detailed comparisons to prior studies to highlight any new insights. To enhance the manuscript, the authors should consider clarifying how the current findings offer new contributions beyond what has already been established in their previous work.
Response: Differing from previous studies, please check the above response to the similar question.
In contrast to previous works, this chapter discusses the significant role of the surface structure of tea dhools in the adsorption of aromatic volatiles, along with potential molecular mechanisms, thereby enhancing the novelty of the article.
Comments on the Quality of English Language
The English and writing can be improved, especially the connections between ideas, paragraphs, and sections.
Response: Yes, We have optimized the language logic.

Reviewer 2 Report
Comments and Suggestions for Authors
Line 63: Please elaborate what you mean by processing tea further beyond. I recognize this might be a wording issues.
I recognize that you are normalize your samples to the total area to determine major compounds. However, why are you not using an internal standard especially with SPME.
While I recognize linalool and geraniol are technically alcohols. You should really separate them out further into terpenes. Terpenes make up such a large class of compounds in teas.
*Please check your spelling for alpha-terpinene.
Author Response
Point-by-point responses to reviewers' comments
Reviewers' comments:
Comments and Suggestions for Authors
Line 63: Please elaborate what you mean by processing tea further beyond. I recognize this might be a wording issues.
Response: Yes, we have corrected it. Thanks a lot for pointing out it.
I recognize that you are normalize your samples to the total area to determine major compounds. However, why are you not using an internal standard especially with SPME.
Response: In our preliminary studies, we used ethyl decylate as an internal standard for quantitative determination of the volatiles, but poor repeatability and stability of its peak area were observed. Therefore, the internal standard method was not used in this study.
While I recognize linalool and geraniol are technically alcohols. You should really separate them out further into terpenes. Terpenes make up such a large class of compounds in teas.
Response: Previous studies have mostly classified compounds linalool and geraniol as alcohols, and their flavor attributes are significantly distinct from those of most terpenes. Therefore, we similarly categorize them as alcohols.
*Please check your spelling for alpha-terpinene.
Response: Yes, we have already checked it.

Round 2
Reviewer 1 Report
Comments and Suggestions for Authors
The authors have addressed the majority of the comments raised in the previous review. The revisions made have improved the clarity and overall quality of the manuscript.
For the aspects that were not modified, the explanation provided regarding the lack of sensory evaluation, due to limitations in technical personnel and equipment, is acknowledged and considered a reasonable justification.
In general, the manuscript is now more robust and better structured. No further major concerns are noted at this stage.
Author Response
Comments:
The authors have addressed the majority of the comments raised in the previous review. The revisions made have improved the clarity and overall quality of the manuscript.
For the aspects that were not modified, the explanation provided regarding the lack of sensory evaluation, due to limitations in technical personnel and equipment, is acknowledged and considered a reasonable justification.
In general, the manuscript is now more robust and better structured. No further major concerns are noted at this stage.
Response: Yes, thank you for your feedback. It will greatly help improve the quality of our manuscript.
Reviewer 2 Report
Comments and Suggestions for Authors
The changes made to the manuscript have greatly improved the paper.
Author Response
Comments: The changes made to the manuscript have greatly improved the paper.
Response: Yes, thank you for your feedback. It will greatly help improve the quality of our manuscript.